# Displacement of Hospital-Acquired, Methicillin-Resistant *Staphylococcus aureus* Clones by Heterogeneous Community Strains in Kenya over a 13-Year Period

**DOI:** 10.3390/microorganisms12061171

**Published:** 2024-06-08

**Authors:** Justin Nyasinga, Zubair Munshi, Collins Kigen, Andrew Nyerere, Lillian Musila, Andrew Whitelaw, Wilma Ziebuhr, Gunturu Revathi

**Affiliations:** 1Department of Pathology, Aga Khan University, Nairobi P.O. Box 30270-00100, Kenya; justinnyasinga@gmail.com (J.N.); mmunshi54@gmail.com (Z.M.); 2Department of Biomedical Sciences and Technology, Technical University of Kenya, Nairobi P.O. Box 52428-00200, Kenya; 3Institute of Science, Technology & Innovation, Pan-African University, Nairobi P.O. Box 62000-00200, Kenya; knyerere@jkuat.ac.ke; 4Walter Reed Army Institute of Research—Africa, Kericho P.O. Box 1357-20200, Kenya; ckigen.ck@gmail.com (C.K.); lillian.musila@usamru-k.org (L.M.); 5Division of Medical Microbiology and Immunology, Stellenbosch University, Matieland, Stellenbosch 7602, South Africa; awhitelaw@sun.ac.za; 6Institute of Molecular Infection Biology, Josef-Schneider Str. 2D/15, D-97080 Wurzburg, Germany; w.ziebuhr@mail.uni-wuerzburg.de

**Keywords:** *Staphylococcus aureus*, MRSA, antibiotic resistance, AMR, epidemiology, genomic surveillance, Kenya

## Abstract

We determined antibiotic susceptibility and employed Oxford Nanopore whole-genome sequencing to explore strain diversity, resistance, and virulence gene carriage among methicillin-resistant *Staphylococcus aureus* (MRSA) strains from different infection sites and timepoints in a tertiary Kenyan hospital. Ninety-six nonduplicate clinical isolates recovered between 2010 and 2023, identified and tested for antibiotic susceptibility on the VITEK ID/AST platform, were sequenced. Molecular typing, antibiotic resistance, and virulence determinant screening were performed using the relevant bioinformatics tools. The strains, alongside those from previous studies, were stratified into two periods covering 2010–2017 and 2018–2023 and comparisons were made. Mirroring phenotypic profiles, *aac(6*′*)-aph(2*″*)* [aminoglycosides]; *gyrA* (S84L) and *grlA* (S80Y) [fluoroquinolones]; *dfrG* [anti-folates]; and *tet(K)* [tetracycline] resistance determinants dominated the collection. While the proportion of ST239/241-t037-SCCmec III among MRSA reduced from 37.7% to 0% over the investigated period, ST4803-t1476-SCCmec IV and ST152-t355-SCCmec IV were pre-eminent. The prevalence of Panton–Valentine leucocidin (PVL) and arginine catabolic mobile element (ACME) genes was 38% (33/87) and 6.8% (6/87), respectively. We observed the displacement of HA-MRSA ST239/241-t037-SCCmec III with the emergence of ST152-t355-SCCmec IV and a greater clonal heterogeneity. The occurrence of PVL+/ACME+ CA-MRSA in recent years warrants further investigations into their role in the CA-MRSA virulence landscape, in a setting of high PVL prevalence.

## 1. Introduction

*Staphylococcus aureus* is a bacterial pathogen with a high clinical burden across the globe, owing to its ability to cause severe infections and resist antimicrobial chemotherapy [1]. *S. aureus* may cause infections ranging from superficial and mild skin infections to severe pneumonia, sepsis, osteomyelitis, and endocarditis. Methicillin-resistant *S. aureus* (MRSA) strains remain the hallmark of *S. aureus* multidrug resistance (MDR). Importantly, MRSA strains have attained and sustained their global presence within both healthcare (HA-MRSA) and community (CA-MRSA) settings, making *S. aureus* a WHO priority pathogen for surveillance and new drug discovery [2]. Livestock-adapted MRSA (LA-MRSA) strains present an additional challenge, especially in Europe and America, with recent evidence of zoonotic transmission [3].

The co-evolution of virulence and resistance with the spread of CA-MRSA strains encoding virulence genes, Panton–Valentine leucocidin (PVL), arginine catabolic mobile element (ACME), and diverse enterotoxins, necessitate epidemiological vigilance. Multi-locus sequence typing (MLST), which indexes polymorphisms in seven housekeeping gene fragments, is a widely used method for studying the evolution of *S. aureus.* Strains sharing alleles in all loci belong to the same sequence type (ST), while STs sharing at least five alleles belong to larger groupings called clonal complexes (CCs). ACME is a genomic island that has been attributed to the global success of USA300 [ST8, SCC*mec* IVa(2B), t008, and PVL^+^] [4], while PVL is a bicomponent leucocidin that has been linked to severe skin and soft tissue infections and necrotizing pneumonia [5]. Genetic differences have been attributed to the transmissibility and persistence of certain strains in distinct geographical settings [6]. Examples include ST80 in Europe, North Africa, and the Middle East; ST612 in South Africa; ST93 in Australia; ST59 in Asia; and ST88 in some parts of Africa [7]. This notwithstanding, the genetic structure of MRSA in a given setting is in a state of flux, with newer, fitter strains displacing established ones over time. This is exemplified by the replacement of ST5 with ST8 in North America, ST239 with ST59 in China [8], ST239 with ST22 in Singapore [9], and EMRSA-16 (ST36) with EMRSA-15 (ST22) in the United Kingdom [10]. A 2015 systematic analysis depicted African MRSA strains as, generally, being structured along clonal complexes (CCs) CC5, CC8, CC22, CC30, CC80, and CC88, with CC5 being prevalent [11].

In Kenya, reported MRSA rates have varied between 3.7% in two private hospitals [12], 27.8% in Kenya’s largest national referral hospital [13], and 53.4% from the same facility [14]. A number of studies have reported on the molecular characterization of MRSA in Kenya, where the predominance of HA-MRSA strain ST239/241-t037-SCC*mec* III(3A), especially among inpatients, has been observed, with other strains being infrequently encountered [15,16,17]. However, an understanding of MRSA dynamics in different geographical, temporal, and clinical contexts remains incomplete. We applied whole-genome sequencing to gain insights into the population structure, antibiotic resistance, and virulence gene carriage among 96 Kenyan MRSA isolates from different clinical, demographic, and temporal backgrounds.

## 2. Materials and Methods

### 2.1. Patient Population

Ninety-six archived, nonduplicate clinical MRSA strains spanning a 13-year period (2010–2023) were randomly selected for characterization using whole-genome sequencing. These strains were recovered from patients seeking inpatient and outpatient services at Aga Khan University Hospital, Nairobi and its countrywide network of outpatient clinics. Clinical and demographic information for the patients, such as age, sex, specimen type, hospital location, geographical origin, and hospital admission status, were retrieved from the hospital laboratory information system (LIS) and electronic health records (EHRs) system. 

### 2.2. MRSA Strain Collection

The MRSA strains were identified using a VITEK^®^ GP ID card and antimicrobial susceptibility testing (AST) was performed using a VITEK^®^ P580 AST card (Biomérieux, Marcy-l’Étoile, France). The isolates were preserved in brain heart infusion broth with 15% *v*/*v* glycerol at −80 °C. For confirmation of purity and MRSA phenotype, the isolates were revived on a blood agar base (Oxoid, Cheshire, UK) with 5% sheep blood and were screened using a 30 µg cefoxitin disc (Oxoid, Cheshire, UK) on Mueller–Hinton agar (Oxoid, Cheshire, UK), where a cutoff of ≤21 mm was used to infer methicillin resistance, as per the guidelines of the 31st edition of the Clinical and Laboratory Standards Institute [18].

### 2.3. Nucleic Acid Extraction

DNA was extracted using a Quick-DNA Fungal/Bacterial Miniprep Kit (Zymo Research, Irvine, CA, USA), following the manufacturer’s recommendations, with the only modification being an increase in the cell disruption time on the thermomixer compact (Sigma Aldrich, St. Louis, MO, USA) for 15 min. The purity of the DNA extracts was determined using a NanoDrop spectrophotometer (Thermo Fisher Scientific, Waltham, MA, USA), followed by quantification using a Quantus™ fluorometer (Promega, Madison, WI, USA). DNA extracts were stored at −20 °C until further processing.

### 2.4. ONT Sequencing, Quality Control Check, and Bioinformatics Analysis

DNA sequencing libraries were prepared based on Oxford Nanopore Technologies (ONT) SQK-LSK-112 chemistry, following manufacturer-provided/recommended kits and protocols. Sequencing libraries were loaded onto an R10.3 flow cell and sequenced on an Mk1B MinIon device (ONT) for 48 h. Raw data were acquired using MinKnow Software (Release 22.08.4) and demultiplexing and quality control checks were performed using ONT’s Guppy Basecaller. Sequences were assembled with Flye 2.9.1 [19] and were polished with Medaka Version 1.7.3. Finally, assemblies were visualized using the Bioinformatics Application for Navigating De Novo Assembly Graphs Easily (BANDAGE) v0.9.0 [20]. Sequences with incomplete assemblies and/or those with sequencing depths of less than 30× were dropped from further analysis, based on the recommendations of Khrenova et al. [21].

### 2.5. Identification of Resistance and Virulence Determinants

SpeciesFinder version 2.0 [22] and KmerFinder version 3.2 [23] were used for *S. aureus* identity confirmation. Antimicrobial resistance (AMR) determinants (genes and gene mutations that confer tolerance to antimicrobial compounds in bacteria) were identified using ResFinder version 4.1 [24], Kmer Resistance version 2.2 [25], and the Comprehensive Antibiotic Resistance Database (CARD) [26]. In addition, genomes were screened for AMR determinants using the AbritAMR pipeline, which is based on NCBI’s AMRFinderPlus [27]. Virulence determinants were detected on VirulenceFinder version 2.0 [28]. All searches were based on default parameters.

### 2.6. Molecular Typing

Staphylococcal cassette chromosome *mec* (SCC*mec*) typing with SCC*mec*Finder version 1.2; Multilocus sequence typing (MLST) with MLST version 2.0 [29] and staphylococcal protein A *(spa)* typing based on SpaTyper version 1.0 [30], all contained within the Centre for Genomic Epidemiology (CGE) platform, were performed, based on the default parameters of the respective programs. To cluster *spa* types, tandem repeat orders were used to retrieve tandem repeat nucleotide sequences from the SpaServer database (https://spa.ridom.de/ accessed on 12 July 2023), which were then concatenated and clustered using the Based Upon the Repeat Pattern (BURP) algorithm using Ridom StaphType Software version 2.0 (Ridom Bioinformatics). Clustering parameters were set as “exclude *spa* types with <5 repeats and cluster sequences if cost ≤ 4” parameters [31]. Nomenclature for SCC*mec* elements was based on the guidelines of the International Working Group on the Classification of Staphylococcal Cassette Chromosome Elements [32]. Clustering of MLST STs was based on the Based Upon Related Sequence Type (BURST) algorithm contained within the PubMLST database.

For insights into temporal changes in strain diversity, we compiled MRSA strain type data from this study (n = 86) and from previous studies by Aiken et al. (n = 6) [15], Omuse et al. (n = 24) [17], Kyany’a et al. (n = 8) [16], Nyasinga et al. (n = 6) [33], Njenga et al. (n = 4) [34], and Obanda et al. (n = 2) [35]. The combined set constituted 136 strains that were split into two periods covering the years 2010–2017 (n = 77) and 2018–2023 (n = 59). For phylogenetic analysis of STs, concatenated sequences of the allelic profiles of individual STs from this combined dataset were generated from the PubMLST website (https://pubmlst.org/) [36]. Multiple sequence alignment was performed with Multiple Alignment using Fast Fourier Transform (MAFFT) and MUSCLE. A midpoint rooted, maximum likelihood (ML) tree with a generalized time-reversible (GTR) model was constructed using Molecular Evolutionary Genetic Analysis (MEGA) Software Version 11.0.11, based on 1000 bootstrap iterations. The tree was refined using the Interactive Tree of Life (iTOL) Version 5.

## 3. Results

### 3.1. Clinical and Demographic Characteristics of the Study Population

The 96 patients had mean and median ages of 37.5 yrs and 38.5 yrs, respectively, with the majority (62.5%) being males. The inpatient and outpatient representations were 54% and 46%, respectively. Skin and soft tissue, respiratory, and blood specimens were the most common sources of MRSA and approximately 72% of the patients were from Nairobi County (Table 1).

### 3.2. Antimicrobial Susceptibility Patterns

All 96 strains showed phenotypic resistance to penicillin, cefoxitin, and oxacillin. The last-line agents tigecycline, linezolid, vancomycin, and teicoplanin all elicited 100% susceptibility, followed by high susceptibility to rifampicin (96%) and clindamycin (82%). Isolates showed progressively declining susceptibilities to the other agents tested (Figure 1).

Seventy-nine percent (76/96) of the strains showed a multidrug resistance (MDR) phenotype (defined as resistance to three or more antimicrobial classes), ranging from three to eight classes. There were 21 MDR combinations, with the most common being “methicillin-gentamicin-ciprofloxacin-tetracycline-TMP-SXT”, which occurred in 15 isolates. Sixty-two percent (47/76) of MDR strains showed resistance to between three and four drug classes. Among the 20 non-MDR strains, eight were resistant to methicillin only, and twelve were resistant to two drug classes. Figure 2 summarizes the resistogram for 76 multidrug-resistant MRSA strains.

### 3.3. Antimicrobial Resistance Determinants

Of the 96 genomes analyzed, 87 were complete, while 9 were dropped because of incompleteness or lower sequencing depths. Whole-genome sequencing coverage (the number of times the whole genome was sequenced) ranged from 30× to 96×, with an average depth of 36×. AMR gene carriage per genome varied between two and thirteen genes. In general, aminoglycoside resistance genes showed the greatest diversity, with the bifunctional acetyltransferase/phosphotransferase *aac(6*′*)-aph(2*″*)* dominating. Further, *aac(6*′*)-aph(2*″*)* was present in all cases of phenotypic resistance to gentamicin. Other aminoglycoside resistance genes occurred in combination, whereby *aph(3*′*)-III* was common. Low-affinity *dfrG* was the most frequently encountered anti-folate resistance determinant (*dfrG*, n = 37; *dfrB*, n = 2; *dfrK*, n = 1), while the tetracycline efflux pump *tetK* was dominant (*tetK*, n = 35; *tetM*, n = 12). Macrolide-lincosamide-streptogramin_B_ (MLS_B_) resistance-conferring methylase genes *ermA* and *ermC* were rarely encountered, as shown in Figure 3. MS_B_ resistance genes *msrA* and *mphC* were encountered equally frequently. Among mutations conferring fluoroquinolone resistance in the quinolone resistance-determining region (QRDR), the *gyrA* serine-leucine substitution (S84L)/*grlA* serine-tyrosine substitution (S80Y) combination of mutations was the most frequent. Consistent with a lack of phenotypic resistance, there were no determinants suggestive of resistance against vancomycin, teicoplanin, linezolid, or tigecycline.

There was a 97% phenotype–genotype concordance, with mismatches involving fluoroquinolones, trimethoprim, fusidic acid, mupirocin, and rifampicin. In the absence of phenotypic resistance against fusidic acid and mupirocin, *fusA* proline-glutamine [P404Q]; histidine-tyrosine [H457Y]; serine-phenylalanine [S416F]; and leucine-phenylalanine [L461F] substitutions (n = 5), as well as the *mupA* gene (n = 2) were detected. All but one of the mismatches involved the presence of resistance determinants without phenotypic resistance. The exception was phenotypic resistance to fluoroquinolones without QRDR mutations, possibly due to overexpression of chromosomal major facilitator superfamily (MFS) fluoroquinolone and biocide multidrug efflux pumps such as *norA*, *norC*, and *sdrM*, which were detected in the genome of the isolate. Multidrug efflux pumps and other antibiotic- and biocide-resistance determinants are shown in Appendix A.

### 3.4. Strain Diversity

SCC*mec* IV(2B) and SCC*mec* V(5C) represented 73% (63/86) and 15% (13/86) of all strains, respectively. SCC*mec* III(3A) (n = 9) and SCC*mec* II(2A) (n = 1) were rare. There were six and three subtypes for SCC*mec* IV(2B) and SCC*mec* V(5C), respectively, and SCC*mec* IVa(2B) dominated (n = 36). One strain of MRSA could not be typed, even though the *mecA* gene was detected with the closest homology (55.29%) to SCC*mec* IV(2B&5). The distribution of SCC*mec* elements is summarized in Table 2. In total, 31 *spa* types were identified, among which t1476 (n = 19), t355 (n = 13), and t037 (n = 6) were common. BURP clustering of the *spa* types generated seven clusters, as follows: *spa* CC121, *spa* CC355, *spa* CC442, *spa* CC005, and three clusters with no founders. Eight *spa* types, represented by eleven strains, were classified as singletons, as shown in Figure 4. Overall, *spa* CC121 (n = 34) and *spa* CC355 (n = 15) were the most common. For seven MRSA genomes, *spa* types were not identified.

Classification using MLST generated 19 sequence types (STs), where ST4803 (n = 30), ST152 (n = 12), and ST7894 (n = 6) were the most common. The remaining 16 STs were infrequently detected. A unique allelic profile [*arcC484*, *aroE1*, *glpF1*, *gmk1*, *pta1*, *tpi1,* and *yqil1*] was submitted to PubMLST [36] for curation and was assigned ST8511 and CC1. BURST clustering grouped 52.3% (45/86) of the strains into the MLST clonal complex (CC) CC8, with CC5 (n = 7), CC30 (n = 5), CC22 (n = 3), and CC1 (n = 2) occurring in declining frequencies. Twenty-eight percent (24/86) of the strains were identified as singletons. One strain was not definitively typed using MLST, despite its genome meeting the recommended inclusion criteria for post-assembly analysis (54×).

### 3.5. Temporal Changes in MRSA Strain Diversity

Over the two periods (2010–2017 and 2018–2023), ST239/241 proportions among MRSA strains declined from 37.7% (29/77) to 0; ST152 increased from 2.5% (2/77) to 20% (12/59); and ST4803 increased from 11.7% (9/77) to 36% (21/59). The 30 unique STs in the dataset showed limited overlaps (9/30) between the two periods. There was a greater diversity of MRSA strains within the globally successful MLST CCs—CC5, CC8, CC22, and CC30. Approximately 60% (82/136) of the strains clustered under CC8 (Table 3). The MLST phylogenetic analysis reflected groupings similar to those of PubMLST, except ST6 and ST789, which were classified as CCs 5 and 8, respectively, but were grouped differently in the phylogenetic tree (Appendix A). Similar to ST239/241, SCC*mec* III strains declined from 44.7% (38/85) to 3.3% (2/60) (Table 4).

### 3.6. Panton–Valentine Leucocidin (PVL) and Arginine Catabolic Mobile Element (ACME) Genes

The virulence of CA-MRSA has been linked to the expression of the PVL and ACME genes. Given the abundance of CA-MRSA strains in our collection, we explored the carriage of PVL and ACME genes. PVL was present in 38% (33/87) of MRSA strains. All PVL^+^ strains carried either SCC*mec* IV(2B) or SCC*mec* V(5C) elements. All strains belonging to *spa* type t1476 (the most common *spa* type) were PVL-negative. ACME was observed in 6.9% (6/87) of MRSA strains. Interestingly, all ACME-positive strains were PVL positive and all carried the SCC*mec* IVa(2B) element. The strains were recovered between 2018 and 2023, where five were from inpatients in the ICU, cardiothoracic ICU, and surgical wards and one was from an outpatient source. A full toxin gene summary is provided in Appendix A.

## 4. Discussion

### 4.1. Antimicrobial Resistance Patterns of MRSA

We observed moderate MDR rates with full susceptibility to last-line agents in a collection that was primarily composed of SCC*mec* IV(2B) and V(5C). The six strains with broad resistance to seven or eight drug classes carried the SCC*mec* III(3A) element. The predominance of CA-MRSA, with low-level resistance against erythromycin, clindamycin, ciprofloxacin, rifampicin, and gentamicin, has been reported elsewhere [5,37,38]. Consistent with the observed phenotypes, *aac(6*′*)-aph(2*″*)*; *gyrA* S84L and grlA S80Y; *dfrG*, *mphC*, *msrA* and *ermA*; H481N; and *tetK* were prevalent in their respective antimicrobial classes. Similar patterns for these determinants among MRSA strains within the country [16,39] and in the neighboring Tanzania [40] have been reported.

The regional and global distributions of some resistance determinants may vary. For example, *tetM* was rare in our setting (n = 12), but it is the dominant genotype in China [41]. In Iran, prevalences of 32.4% for *tetM*, 17.2% for *tetK*, and 13.9% for *tetK + tetM* were observed [42]. The cooccurrence of *tetK* and *tetM* was infrequent in our study (n = 5). *dfrG* is dominant among Kenyan strains [35]; but, in the UK, *dfrA* is common [43]. The global distribution of *erm* genes is variable, with isolates from certain regions showing higher frequencies of *ermA* (South America), *ermB* (China), and *ermC* (Europe and middle East) [44]. The few phenotype–genotype discordant results we observed have been acknowledged by others. Tanzanian and Congolese studies noted higher proportions (>60%) of *dfrG* without phenotypic resistance [40,45]. Up to 87% of rifampicin-resistant MRSA carry the H481N mutation, with or without other mutations, whose co-expression may confer high-level rifampicin resistance, but H481N alone may not generate phenotypic resistance [41].

Although in vitro susceptibility to last-line agents has been confirmed by multiple local and regional observations, the recent report of chloramphenicol-florfenicol resistance (*cfr*) gene-mediated linezolid resistance delivers a reminder for greater vigilance [34]. While vancomycin is the drug of choice for MRSA infections, the high prevalence of CA-MRSA with moderate MDR opens the possibility of deploying antibiotics such as clindamycin and rifampicin, especially in settings where isolated AST is routinely performed [46].

### 4.2. Molecular Epidemiology of MRSA in Kenya

We observed the supplantation of the hitherto dominant HA-MRSA clone ST239/241-SCC*mec* III(3A). This decline coincided with the emergence of SCC*mec* IV(2B)- and SCC*mec* V(5C)-bearing strains such as ST152-t355 and ST4803-t1476 in a heterogeneous CA-MRSA population. ST239/241-t37 MRSA strains have been previously reported in Ghana (n = 1), Egypt (n = 2), South Africa, and Ethiopia [7,47] and constituted 40% of MRSA from five African countries by 2008 [48]. The MRSA population structure in Africa has demonstrated temporal shifts, where HA-MRSA strains have declined, while CA-MRSA strains ST22 and ST152 have emerged [7]. In China, ST239-t037 proportions dramatically dropped from 18.5% to 0.5% in the 2008–2017 period, which reflects the global disappearance of this clone [49]. The evolutionary events leading to the initial rise, decades-long dominance, and the eventual fall of ST239 have been linked to the fitness costs of the ~600 kb ST8/ST30 recombination that generated it [50].

ST4803-t1476 was the dominant CA-MRSA in our setting. Similarly, t1476 was preponderant in Congo [45] and Tanzania [40]. MRSA and MSSA strains bearing t1476 have only been rarely (n = 2) observed in Kenya before [17]. In South Africa, t1476 prevailed among pediatric atopic dermatitis patients in rural and urban settings [51]. Although rare, 32 cases of MRSA bearing t1476 with SCC*mec* V(5C) were reported among elderly patients in the UK [52], reflecting a global presence for this clone. In contrast, a study of human and livestock MRSA in Uganda failed to identify t1476 or t355 MRSA, which were the pre-eminent strains in our study [53]. All t1476 MRSA strains in this study and from Tanzania were PVL-negative and carried SCC*mec* IV(2B&5) [40], whereas those from Congo carried SCC*mec* V(5C), suggesting independent SCC*mec* acquisition.

Another prevalent clone was ST152-t355. Unlike t1476, all t355-positive strains carried the PVL gene. The role of PVL in the evolving epidemiology of MRSA has long been raised, as both PVL and *mecA* are borne on mobilizable genetic elements [11,54]. A high PVL prevalence among ST152-t355 MSSA in Africa has been noted [55]. Increasing detection of the PVL^+^/SCC*mec*^+^ ST152-t355 strain suggests the acquisition of SCC*mec* elements by ST152-t355 MSSA. Since all ST152-t355 MRSA strains in this study and those in the study by Kyany’a et al. [16] carried the SCC*mec* IVa(2B) element, a single SCC*mec* acquisition event, followed by successful transmission, rather than multiple independent acquisitions of the SCC*mec* element, may be hypothesized. Outside of Africa, ST152 is common only in the former Yugoslavian countries, where ST152-t355-SCC*mec* V(5C) strains dominate [56]. ST152 MRSA strains containing other *spa* types (t5691 and t15644) and SCC*mec* II, V, and VII have been reported in Congo. It has recently been proposed that ST152, like ST80, originated from Africa in the 1970s and subsequently acquired SCC*mec* elements in Europe in the 1990s [57]. In light of its growing epidemiological significance in Africa, studies assessing ST152 strains from varied spatial and temporal backgrounds, especially in the context of φSa2 prophage and the SCC*mec* element, are needed.

Two lineages seem to lead the race for CA-MRSA dominance in our setting—PVL-negative ST4803-t1476 and PVL-positive ST152-t355. While we can hypothesize that under antibiotic pressure, PVL-positive ST152 MRSA strains will outcompete their PVL-positive ST152 MSSA counterparts, the same cannot be said of PVL-negative ST4803 clones, which seem to succeed in the same setting. Further studies on the relative fitness of these two lineages are warranted. Though MRSA strains with smaller SCC*mec* cassettes such as SCC*mec* IV(2B) and SCC*mec* V(5C) have been observed in various settings in Africa, there does not seem to be any clinical distinctions between HA- and CA-MRSA infections [11].

### 4.3. PVL and ACME Gene Carriage

The prevalence of PVL in Kenya has been reported as 17% among SSTIs [58], 17% among inpatient carriage screens [15], 33% among carriage and invasive strains [59], and 42.9% among abattoir workers [39]. Higher prevalences of 38.5% in Ghana [55], 59.3% in Ethiopia [47], 49% in Congo [45], 61.4% in Gambia [60], and 75% in Egypt [38] have been published. Our study’s 38% prevalence is consistent with a recent continental median estimate of 33% [7]. Outside of Africa, PVL prevalence is typically less than 5% [61,62,63].

We observed an ACME and PVL prevalence of 6.2%, with all strains showing >99% homology to the ACME I element [4]. In Japan, an ACME and PVL positivity rate of 1% has been reported [61]. In Ireland and the Netherlands, ACME prevalences of 9.7% [64] and 8.47% [4], respectively, were noted. Our results (6.2%) compare well with these rates, but contrast with the 47.2% estimate from Iraq [63]. The role of ACME in enhancing bacterial spread and fitness is controversial, with contradicting findings in in vitro and in vivo experiments [5,65]. Others have observed that PVL and/or ACME may not be the only or even prominent determinants of the success of CA-MRSA, but rather players in a complex genetic network [5]. Given the high prevalence of PVL in Africa, research into the implications of these genes in MRSA spread and pathogenicity is overdue.

## 5. Conclusions

Our study had some limitations. As over 70% of the strains originated from the same geographical setting (Nairobi County), we could not perform an in-depth examination of the geographical distribution of MRSA strains in Kenya. In addition, the strains originated from a single, private health facility. However, these data form a solid basis upon which strains from other parts of the country may be compared in the future. There were a few instances where strain types could not be identified, which may point to methodological limitations of the sequencing technology and chemistry used.

In conclusion, we observed moderate MDR rates among MRSA strains alongside commonly implicated resistance determinants. We demonstrated the decline of HA-MRSA ST239/241 with the emergence of CA-MRSA strains such as ST152 and a greater heterogeneity that reflects MRSA’s changing population structure in Africa. Last, we have observed the co-occurrence of PVL and ACME, whose roles in the virulence landscape of MRSA in Africa should be explored.

## Figures and Tables

**Figure 1 microorganisms-12-01171-f001:**
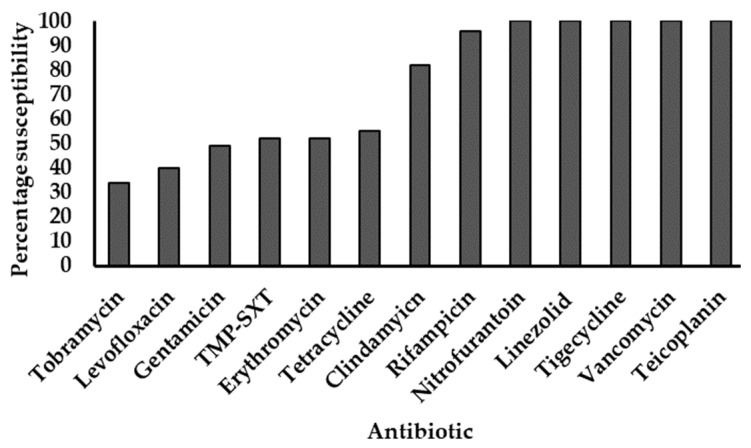
Antimicrobial susceptibility profiles of MRSA strains. TMP-SXT: Trimethoprim-sulfamethoxazole. All strains were resistant to penicillin, cefoxitin, and oxacillin. The *y*-axis represents the proportion of strains susceptible to individual antibiotics (*x*-axis), expressed as percentages.

**Figure 2 microorganisms-12-01171-f002:**
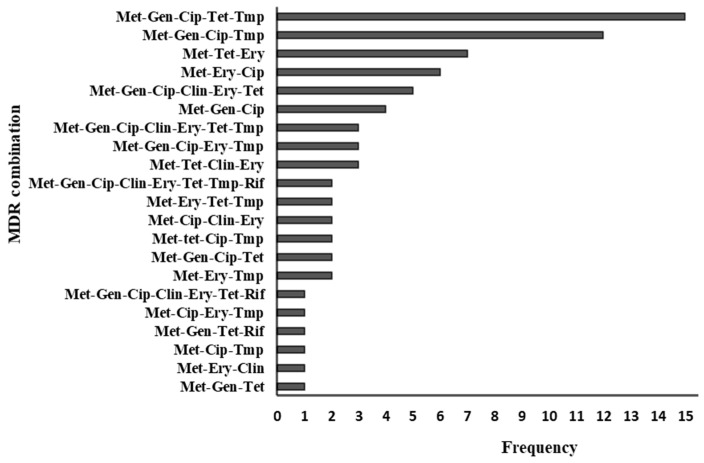
Multidrug resistance patterns of 76 MRSA strains. Met—methicillin, Gen—gentamicin, Cip—ciprofloxacin, Ery—erythromycin, Tet—tetracycline, Clin—clindamycin, Tmp—trimethoprim-sulfamethoxazole, Rif—rifampicin.

**Figure 3 microorganisms-12-01171-f003:**
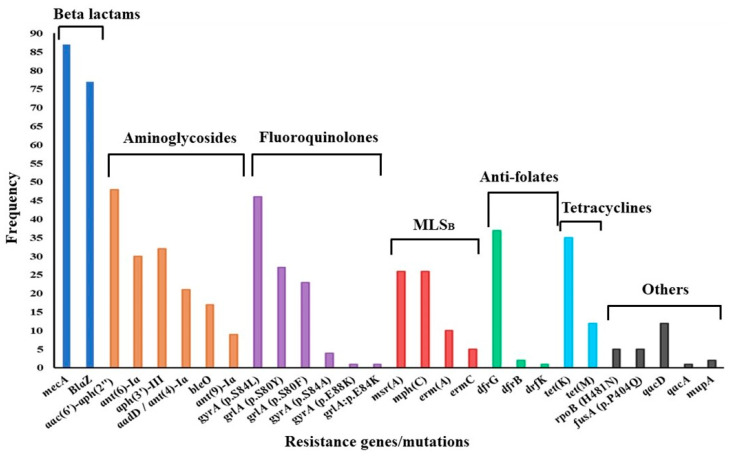
Distribution of antimicrobial resistance determinants among MRSA strains. MLS_B_—macrolide-lincosamide streptogramin B.

**Figure 4 microorganisms-12-01171-f004:**
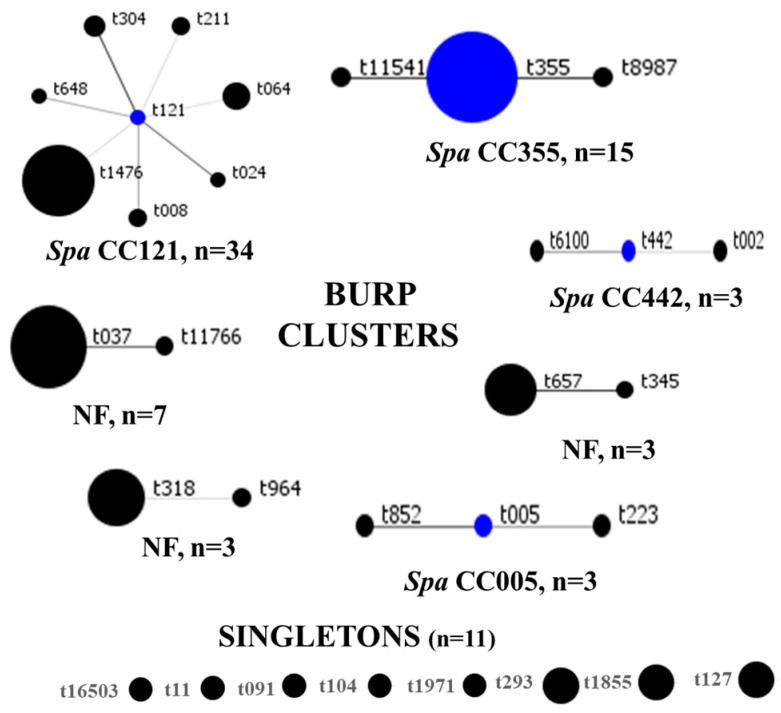
BURP clustering of MRSA *spa* types. Blue circles denote the founder *spa* types, while the size of each circle is proportional to strain frequency. CC: clonal complex, NF: clusters with no founders.

**Table 1 microorganisms-12-01171-t001:** Clinical and demographic characteristics of patients.

Variable	Indicator	Value
Age	Mean [SD]	37.5 [±25.8] yrs
	Median [IQR]	38.5 [15.8–61] yrs
Sex	Male	62.5% (60/96)
	Female	37.5% (36/96)
Admission status	Inpatient	54% (52/96)
	Outpatient	46% (44/96)
Specimens	SSTI	26% (25/96)
	Respiratory	25% (24/96)
	Blood	14.5% (14/96)
	Tissue	11.5% (11/96)
	Nasal carriage	11.5% (11/96)
	Urine	5.5% (5/96)
	Vaginal swab	3% (3/96)
	Miscellaneous	3% (3/96)
Year of isolation	2010–2017	37.5% (36/96)
	2018–2023	62.5% (60/96)
Geographical origin	Nairobi County	72% (69/96)
	Kiambu County	10.4% (10/96)
	Machakos County	4% (4/96)
	Others	13.6% (13/96)

IQR: interquartile range, SD: standard deviation, SSTI: skin and soft tissue infections.

**Table 2 microorganisms-12-01171-t002:** Distribution of SCC*mec* types among MRSA strains.

SCC*mec* Type	SCC*mec* Subtype	Frequency
SCC*mec* II(2A)	*	1
SCC*mec* III(3A)	*	9
SCC*mec* IV(2B)	SCC*mec* IVc(2B)	2
	SCC*mec* IVd(2B)	2
	SCC*mec* IVg(2B)	2
	SCC*mec* IV(2B)	4
	SCC*mec* IV(2B&5)	17
	SCC*mec* IVa(2B)	36
SCC*mec* V(5C)	SCC*mec* Vc(5C2&5)	1
	SCC*mec* V(5C2)	4
	SCC*mec* V(5C2&5)	8

* No subtypes are available for the class of SCC*mec* elements.

**Table 3 microorganisms-12-01171-t003:** Temporal changes in strain diversity for 136 MRSA strains.

Clonal Complex	Sequence Type	*spa* Type	SCC*mec* Type	2010–2017		2018–2023		Total
				This Study (n = 31)	Others (n = 46)	This Study (n = 55)	Others (n = 4)	
CC1 (n = 2)	ST8511 *	t127	IV	–	–	2	–	2
CC5 (n = 8)	ST6 ^	t304/t648	IV	1	–	3	–	4
	ST5	t13150	II	–	1	–	–	1
	ST4166 ^	t442/t6100	IV	1	–	1	–	2
	ST7895	t002	II	–	–	1	–	1
CC8 (n = 82)	ST789	t091	V	–	1	–	–	1
	ST2416	t1476	IV	–	–	2	–	2
	ST4803 ^	t008, t024, t064, t104, t121, t211, t1476	IV, V	9	–	21	–	30
	ST4705	t2029	III	–	1	–	–	1
	ST6610	t293	IV	2	–	–	–	2
	ST7635 ^	t030/t037	III	2	–	1	1	4
	ST7894	t037/t11766	III	6	–	–	–	6
	ST8 ^	t1476/t104	IV	–	4	–	1	5
	ST239 **	t037	III	–	6	–	–	6
	ST241 **	t037/t2029	III, IV	–	23	–	–	23
	ST7460	t1476	IV	–	–	1	1	2
CC22 (n=7)	ST957	t223/t852/t005	IV	3	–	–	–	3
	ST22	t005/t022	IV	–	4	–	–	4
CC30 (n = 9)	ST4789	t964	IV	–	–	1	–	1
	ST4618 ^	t318	IV	2	–	3	–	5
	ST30 ^	t318	IV	–	1	–	1	2
	ST39	t007	II	–	1	–	–	1
Singletons (n = 28)	ST88	t1339	NA	–	1	–	–	1
	ST152 ^	t355/t8987/t11541	IV	–	2	12	–	14
	ST1633 ^	t355	IV	2	–	1	–	3
	ST2167	t1855/t16503	IV	–	–	4	–	4
	ST7670	t657/t345	V	3	–	–	–	3
	ST4440	t091	V	–	–	1	–	1
	ST753	t011	V	–	–	1	–	1
	ST140	NA	IV	–	1	–	–	1

*—Novel sequence type; **—STs commonly associated with HA-MRSA infections; ^—strains showing overlaps between the two periods compared. NA—data were not provided.

**Table 4 microorganisms-12-01171-t004:** Proportions of SCC*mec* III (3A) elements over time.

Period	Total MRSA	No. of SCC*mec* III (3A) ^a,b^	SCC*mec* III (3A) (%)	Reference
2010–13	32	14	44	[17]
2011	6	6	100	[15]
2015	6	4	66	[33]
2015	2	1	50	[35]
2015–17	8	5	63	[16]
2010–17	30	8	27	This study
2018–23	4	1	25	[34]
2018–23	56	1	2	This study

^a^ MRSA strains carrying SCC*mec* III (3A) belonged to either ST239/241, ST7894, or ST7635. ^b^ MRSA strains belonged to *spa* types t030, t037, or t2029.

## Data Availability

The MRSA collection and associated clinical, demographic, and geographic data have been registered on NCBI’s BioProject Database under the BioProject Accession Number PRJNA1046723. The genome sequences have been deposited on NCBI’s GenBank Database and assigned Accession numbers in the range of CP141377–CP141522. All other data generated or analyzed during this study are included in this published article [and its Appendix A Files].

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
