# Peer review of "Displacement of Hospital-Acquired, Methicillin-Resistant Staphylococcus aureus Clones by Heterogeneous Community Strains in Kenya over a 13-Year Period"

_microorganisms, 2024, doi:10.3390/microorganisms12061171_

Round 1
Reviewer 1 Report
Comments and Suggestions for Authors
The manuscript "Displacement of hospital-acquired methicillin-resistant Staphylococcus aureus clones by heterogeneous community strains in Kenya over a 13-year period" addresses a crucial topic in antimicrobial resistance. Understanding the spread of resistant strains, including profiling genomic determinants, is pivotal in addressing the resistance issue. This article provides valuable insights into the genetic determinants of MRSA-resistant strains originating from a private hospital network in Kenya over a 13-year period. The substantial number of sequenced isolates (96 strains in the initial selection) is noteworthy and exceeds that of similar studies. The methods employed are appropriate and well-described, and the manuscript is effectively illustrated and well-written, rendering it suitable for publication in Microorganisms. I have some minor comments on the data, provided below.
Minor comments:
-
1. The attribution of strains isolated from inpatients as hospital-acquired may not be strictly accurate. It's possible that these patients were infected in the community before hospitalization. Clarification on this point would be beneficial.
-
2. The data presented in Figure 1 is not entirely clear. Is this percentage of the total number of strains susceptible to antibiotics? If so, the axis should be labeled accordingly. The term "susceptibility, %" suggests that bars should represent quantitative susceptibility data.
-
3. If available, it would be interesting to include information on the frequency of MRSA isolates among the total number of S. aureus isolates and changes in resistance detection over the studied 13-year period.
Reviewer 2 Report
Comments and Suggestions for Authors
Authors in the manuscript "Displacement of hospital-acquired methicillin-resistant Staphylococcus aureus clones by heterogeneous community strains in Kenya over a 13-year period" perform a descriptive analysis of a collection of S. aureus isolates, including their resistance profile and their evolutionary correlation.
The manuscript is well-written and properly organized. The data presented is complete and supports the conclusion. I have minor comments to improve the paper:
- First of all, I miss some correlation between the phenotypes and genotypes. Besides the article is just descriptive, since the authors have the WGS performed, resistance genes presence and resistance profile, it would be interesting to see the correlation between genotype and phenotype on the strains. For example, according to the different AME detected, which are linked with the resistance to the different aminoglycosides tested?
- Supplemental Figures: quality improvement is needed, and the text should be included as text and not figure. In ST1 the description is cropped, please correct it.
- abstract: gene names in italics (lines 28-34)
- Line 53: necessitate
- Lines 51-66: I recommend to briefly explain the differences between ST and CC
- Lines 67-68: This sentence is confusing since authors mention 2 hospital but 3 percentages.
- Line 101: min
- Line 122: define AMR
- Line 145 and Table 1: In the text authors mention 2 periods, while in the table they are arranged in 3 groups that do not match with the described in the text. Please fix.
- Line 191, 284: in general authors mention words like common or infrequent to relate to presence, could some numbers be provided? Or please specify what are the threshold for each category.
- Line 204: [L461F]
- Line 234: what does 54x mean?
- Figure 4 is a bit pixeled.
- Line 295: do you mean chloramphenicol?
